# Metaphors in Educational Videos

Michele Norscini [1,*] and Linda Daniela [2]

1 Department of Education, Cultural Heritage and Tourism, University of Macerata, 62018 Macerata, Italy
2 Faculty of Education, Psychology and Art, University of Latvia, LV-1586 Riga, Latvia; linda.daniela@lu.lv
* Correspondence: m.norscini1@unimc.it

**Abstract:** Traditionally, metaphors have been used as a pedagogical tool to facilitate the processes of educational mediation. From a medial perspective, there are various ways to implement educational mediation, and currently, we are witnessing an increase in the use of videos. Given the historical pedagogical role of metaphor and the widespread use of videos, we expected to find a significant amount of the scientific literature exploring metaphors in educational videos. However, studies establishing a direct connection are rare. Motivated by this gap, we decided to present users with a metaphorical educational video, intending to observe and analyze, through a phenomenological approach, how the metaphor is perceived by users. To gather data on users' experience, we applied the think-aloud protocol during video consumption and then we conducted semi-structured interviews. Subsequently, we analyzed the collected data using phenomenological procedures. Our results highlighted that the use of metaphor can stimulate engagement and facilitate the educational mediation, as long as the metaphor is shared and perceived as coherent by users. Finally, we have highlighted some distinctive aspects of using metaphor in educational videos, such as the ability to visually represent metaphors, create metaphorical contexts, and reinforce the processes of embodied simulation that occur during video viewing.

**Keywords:** metaphor; educational video; phenomenology; embodied simulation; think-aloud; corporate training; digital learning; multimodal science learning and teaching

## 1. Introduction

Metaphors can be used as a pedagogical tool to facilitate the process of educational mediation. Through them, we can transform the complexity of concepts to be taught into simpler and more familiar ones for the learner [1–5]. As a pedagogical tool, metaphors can be employed in various educational formats, such as written texts, infographics, or educational videos.

In the educational context, particularly within the context of asynchronous corporate training in which this study is situated, we are observing the growing use of the video format to mediate teaching and learning processes. Technological development has played a significant role in the dissemination of this medium; thanks to technology, we can access videos with just a few clicks, whenever and wherever we want. The on-demand nature of videos effectively aligns with the time micro-management characteristic of corporate life, allowing for the scheduling and integration of training moments into busy workdays [6,7]. Corporate academies and faculties have established digital spaces where videos and other asynchronous formats are made available, enabling users to tailor their training based on the specific skills demanded by the current work environment [8].

From a medial perspective, video brings along specific characteristics that manifest in the ability to synchronize and display audiovisual inputs as a stream [9]. Instructional designers leverage this feature to create multimedia learning materials [10]. However, designing an educational video requires more than simply juxtaposing images and words; it necessitates the transformation of expert knowledge into instructional knowledge through a process of didactic transposition [11,12].

Philosophers and scholars have long pondered the best ways to transpose knowledge and make it instructional. Plato, in *Meno*, posited the paradox of learning [1], framing the question as follows: how can we come to know new knowledge? Continuing to explore Plato's dialogues, we observe that Socrates often employs metaphorical strategies to explain complex and abstract concepts such as truth, freedom, or justice to his disciples. The metaphor of the cave in the *Republic* serves as an example of how, through concepts already known and easily imaginable, we can come to understand unknown and complex ideas [4]. Even in the context of corporate training, metaphors are used for instructional purposes; consider, for instance, the theory of the six hats for effective meeting organization [13] or the fishbone diagram for quality management [14].

Given the current widespread use of videos for learning and the historical pedagogical role of metaphor, we expected to find a substantial number of studies investigating the use of metaphor in educational videos. However, we discovered that such studies are rare. Motivated by this gap, we decided to present users with a metaphorical educational video to observe and analyze, using a phenomenological approach, how the metaphor was perceived in user's experience and whether it aided or hindered the learning processes through videos.

## 2. State of the Art

### 2.1. The Metaphor as a Cognitive Tool

Metaphors are commonly described and conceived as a rhetorical tool, primarily used in poetry and arts [5]. They allow artists to describe phenomena through concepts that exceed their literal meaning [15,16]. For instance, Shakespeare's phrase "All the world's a stage" is metaphorical because it employs the concept of the stage to convey a particular image of the world. Normally, the stage and the world are two distinct concepts, but when placed in a metaphorical relationship, their meanings illuminate each other and contribute to highlighting specific aspects that the poet wants to emphasize. Based on this example, we define metaphors as the act of relating two domains that are seemingly distinct from each other, but whose juxtaposition generates a particular meaning [2]. Thanks to their ability, metaphors are also a tool used in art therapy, allowing patients to express the abstract nature of their subjective interiority [17].

Even in scientific language, we observe the use of metaphors. For example, terms like "big bang", "black holes", and "messenger RNA" are metaphorical expressions since they borrow terms and concepts from other domains to describe natural phenomena [1,4,5]. Through such metaphorical expressions, scientific theories are constructed, and it is not possible to distinguish clearly between the use of metaphorical language and scientific language [18]. Other researchers have also delved into analyzing the use of metaphors in everyday language [19,20]. In these studies, it is highlighted that a significant portion of everyday language is metaphorical, and it is emphasized that we often use metaphors unconsciously. Expressions like "I'm in high spirits" or "I'm feeling down" may not be perceived as metaphorical, yet they are because spatial concepts of "high" and "down" are used to express abstract feelings of happiness or sadness.

In the analysis of everyday and scientific language, theories have been put forward regarding the role of metaphor in cognitive processes, asserting that if we express ourselves through metaphors, it is because we conceptualize the real by metaphorizing it. A metaphor is not limited to being merely a poetic tool but becomes the way in which the world is conceptualized [5,19,20]. Metaphors allow us to reify phenomena, to structure them, and to orient ourselves within them through concepts that we already know [20].

### 2.2. The Metaphor as a Pedagogical Tool

In the process of didactic transposition and mediation, metaphor can be used pedagogically to reduce conceptual alterity and stimulate conceptual restructuring [2]. New concepts to be learned, especially abstract and complex concepts, can be explained and understood metaphorically through already-known, simpler, and more concrete concepts [1–5]. For

example, over the centuries, various metaphors have been employed to teach the function-ing of the brain: Leibniz associated the brain with a mill, Freud with an electromagnetic system, and today we associate it with a computer [21]. Based on these metaphors, scientific theories on the brain were then explained and learned. Note that the metaphors chosen to explain–comprehend complex concepts always fit within the cultural context in which they arise [3]. Familiarity with the metaphorical concept, such as 'computer', employed to explain the metaphorized concept, for instance, 'brain', is a key factor in understanding the topic, because, if the metaphor is not understood or is seen as incoherent by the student, learning becomes more complex rather than simplified [2]. Furthermore, each metaphor tends to highlight some aspects of the metaphorized concept while simultaneously hiding others [1,20]. Therefore, choosing the right metaphor for teaching and learning is a delicate operation because it may not be understood or shared and may exclude important elements to be learned. Another aspect to avoid is that the learner identifies the metaphor with its reference; in fact, as emphasized by Cameron [2], there is a risk that learners do not realize the metaphorical sense of the explanation. In general, if a metaphor is used for teaching–learning, it must be recognized by the learner as such. In addition to this, it must be judged coherent with the training objectives, with the concept to which it refers, and with the context in which it fits.

Finally, we underline that using metaphors for teaching–learning does not mean reject-ing logical–analytical thinking but rather promoting and explicating the osmotic processes that are already at work in language and thought. Using metaphors as a pedagogical tool means combining metaphorical explanations, which help us understand complexity, with analytical explanations, which allow us to delve into complexity [1,4].

*2.3. The Metaphor in Educational Videos*

In the literary review by Fyfield et al. [22], that which makes an educational video it is analyzed. Regarding the term "video", it is considered a "self-explanatory" concept, and following the definition provided by Ibrahim et al. [9], the video is described as a format in which information is presented as a dynamic stream of visual and auditory content. The addition of the adjective "educational" before the term "video" transforms it from being informative to being formative. Essentially, what distinguishes an educational video from other types is the presence of an intent to teach and, consequently, an explicit learning goal [22]. Winslett [23] defines educational video as a "meta-genre" that encompasses vari-ous exemplars "structured into 19 types". In the literature, there are multiple taxonomies of educational videos, each differing quantitatively and qualitatively from the others [24–27].

We have analyzed various taxonomies and have not found references to metaphorical or metaphor-based educational videos. Generally, reflections on the role of metaphor within educational videos in the scientific literature are scarce and rare. In the few identified studies, metaphor is viewed as a tool used at certain moments by the teacher, mostly from a discursive perspective. For example, a study by Alnajjar et al. [28] introduces a method for tracking metaphors within the textual content found in the videos. A study by Schabarum and Chishman [5] analyzes the use of metaphors in the verbal discourse of the biology teacher in video lessons. Other studies focus on the use of metaphors in the teacher's discourse during video lessons but including gestures and postures in the analysis [29,30]. The video lessons examined in the cited studies, although using metaphors, do not fall into what we define as metaphorical educational videos. With a metaphorical educational video, we refer to a specific type in which the knowledge to be learned is visually situated in a metaphorical space and it is multimodally expressed in light of the chosen metaphor. With 'metaphorical space', we mean that the video is situated within a metaphorical setting, seemingly unrelated to the discussed topic, but whose connection allows for the emergence of specific meanings. With 'multimodally expressed', we mean that the learning content can be metaphorically conveyed not only from a linguistic perspective but also visually and sonically. Moreover, the metaphor helps to express the learning content in ways different from the conventional methods of direct instruction, adopting, for example, a storytelling

approach in which a metaphorical situation is simulated. In the literature, we have not found case studies or reflections on educational videos of this kind.

However, we emphasize that reflections on the visual role of metaphors and the multimodal expression of content in light of metaphors come from the field of video advertisements [31–34]. In the examined video advertisements, the metaphor is not only a verbal or paraverbal tool used by the speaker at certain moments but is primarily a visual tool that constitutes a metaphorical setting through which the video's message is conveyed. In these studies, we note that the metaphor generates meaning and allows for greater engagement and participation from the viewer. However, as video advertisements, they do not have educational purposes.

*2.4. The Use of Metaphors in the Cognitive Multimedia Learning Theory*

We wondered whether the reason behind the absence of studies on metaphorical educational videos and the rarity of reflections between educational videos and metaphors is due to a possible interpretation of the cognitive load that underlies the Cognitive Multimedia Learning Theory (CMLT), 'the most common theoretical lens used to design and evaluate instructional videos' [22] (p. 155). To design and evaluate videos, we can refer to the list of principles for effective multimedia learning design proposed by Mayer et al. [35]. These principles were provided based on the interpretation of cognitive load theory [36–38] and are grounded on three main assumptions: the dual-channel assumption, the limited capacity assumption, and the active processing assumption [39]. According to CMLT, working memory operates effectively when the cognitive load, both intrinsic and extrinsic, is appropriate. Therefore, one might think that the use of metaphors is an unnecessary and harmful burden on extrinsic cognitive load. In an interesting study conducted by Moreno and Mayer [40], they analyzed the role of metaphor in explaining mathematical concepts through multimedia formats. It is important to note that the investigated formats are not videos, but static slides presented in a series. In this study, they demonstrated that metaphor does not generate an increase in cognitive load; on the contrary, it allows the concept to be explained in a simplified manner, facilitating the learning process and memory function. Additionally, the visual and figurative nature of metaphor can be well expressed through multimedia formats, as they have the capability to reproduce images and synchronize them with text and audio. Therefore, even from the perspective of CMLT, metaphor proves to be a tool that can help break down complexity and enable a multimodal understanding of the concept to be learned.

*2.5. Embodied Simulation in Video Consumption*

Recent neuroscientific research has revealed that when observing the actions of others, a process of embodied simulation takes place within the human brain. This process is multimodal and activates various body systems simultaneously, including the motor system that underlies perceptual, cognitive, and linguistic processes [41–45]. Embodied simulation also occurs during the consumption of videos [46–48]. The actions, words, images, and sounds encountered on the screen are simulated within us. The act of simulation means that we do not recreate the video within us exactly as it appears on the screen. Instead, the perceived video is transformed based on the relationship between our existing knowledge and the observed phenomenon, infusing it with a multimodal corporeal and emotional dimension. Throughout this process, we establish a direct relationship with the video in a dual movement: on one hand, the video enters within us, and on the other, we ourselves enter into the video [46]. The concept of 'embodied simulation' highlights that the relationship between the user and the video is not merely a visual phenomenon but an embodied experience.

## 3. Research Method

### 3.1. Research Context

The research operates within the context of asynchronous and online corporate training. The video analyzed in this study is part of the digital library of an Italian learning company that provides asynchronous online training on the main soft and digital skills required by the current work environment. The library is implemented on an online platform that has participants from Italian and international companies. The investigated educational video, titled "Cyber Gym: The Gym for Becoming Cybersecurity Champions", is currently utilized by workers at all levels and from various sectors to receive training in cybersecurity—a skill that has assumed a crucial role in the corporate landscape in recent years. The video is in the animated cartoon format and the main characters are a personal trainer who is expert in cybersecurity, and a trainee who needs to learn how to protect himself from online risks. In the video, the two characters engage in dialogue, simulating the teaching and learning process. We define the "Cyber Gym" video as a "metaphorical educational video" because cybersecurity concepts are transposed into a metaphorical space, specifically the gym setting. The metaphor of the gym permeates and structures the entire video: the triadic structure of the video—warm-up, training, cool-down—mirrors a typical gym session; the language used, the characters, and the dialogue style simulate the teaching and learning experience typical of the gym. In Appendix A there are some frames of the video.

### 3.2. Research Questions

Considering the novelty of the video genre under investigation and the scarcity of reflections on the use of metaphors in educational videos, we decided to analyze the user experience, adopting a phenomenological approach. This choice aims to explore and understand how users perceive and interpret metaphorical elements in the investigated video. The phenomenological approach is well-suited for this research as it allows us to delve into the subjective experiences of users, capturing the essence of their perceptions without imposing preconceived categories or biases. By embracing phenomenology, we aim to uncover the nuanced and individual ways in which participants engage with metaphorical content in the educational video, providing a rich and contextually grounded analysis.

We investigated the 'enacted object of learning' [49], which is the moment when the user interacts with the educational material designed by the instructional designer—in this case, a metaphorical educational video on cybersecurity—in order to answer the following question:

- How is the metaphor used in the video perceived by the users?

From the users' perceptions, we aim to understand if the use of metaphor either aids or hinders the learning process through videos.

### 3.3. Phenomenological Approach

We adopted the phenomenological approach that allowed us to 'go back to the things themselves' by implementing an attitude of epoché, i.e., suspension of judgment, to observe and analyze the user's experience and their perception of the metaphor in the most impartial manner [50]. Since its inception, the phenomenological approach has been characterized by two streams of thought [51]: the descriptive phenomenological approach [50,52,53] and the interpretative phenomenological approach [54–58]. We chose to integrate reflections and methods from both descriptive and interpretative perspectives, therefore, despite acknowledging that pure observation and description of the phenomenon are impossible [54–58], we made an effort, to the best of our ability, not to manipulate the descriptions of the experience [50,52,53], and we endeavoured to clearly distinguish our interpretations from the descriptions provided by the users. The techniques used for data collection, the methods of analysis, and the results presented are the outcome of this effort.

*3.4. Research Design*

The study utilized a non-representative convenience sample consisting of ten participants who are registered on the online platform where the video is uploaded. Each participant was interviewed individually: in the first phase of the interview, we asked the user to watch the video and speak out loud, expressing their thoughts during the viewing; in the second phase of the interview, after the video consumption, we posed some questions to delve deeper into the user's experience. Following the phenomenological approach, we never informed the user about our interest in the metaphor and never directly asked for their thoughts on the metaphor to avoid manipulating their experience and leading them to provide information they might not have spontaneously shared. The data collected on the gym metaphor were therefore expressed spontaneously. Over approximately one month, we interviewed all individual participants. In the following sections, we describe in detail how we composed the sample, collected the data, and analyzed them.

*3.5. Participants*

The conditions to participate in the study were twofold: not having seen the investigated video and being potential consumers of it. To reach participants meeting these conditions, we sent an email to five Italian companies registered on the platform where the "Cyber Gym" video is available. In this email, we provided a general description of the research project as an analysis of the experience with educational videos and asked people to participate in the study. Only three companies responded, and each provided us with a list of users. From the lists, we then selected individual participants for interviews. Through this selection process, we attempted to mitigate self-selection bias by creating a non-representative but mixed sample in terms of age, role, and work experience. Specifically, we requested 5 junior employees from the first company, 2 senior employees from the second, and 1 senior employee from the third. None of them had previously seen the video, and all were already registered on the platform. Subsequently, we approached two senior instructional experts: a professor from an Italian public university and an instructional designer from an Italian learning provider company, asking if they were interested in participating in the study. We aimed to enrich the sample with professionals interested in the video content while also being experts in instructional design. Both experts accepted. Thus, we assembled a non-representative sample of 10 participants. We chose to limit the analysis to only 10 participants, as recommended by Smith [57] for phenomenological analysis. Although the limited number of participants does not allow us to generalize the results regarding user categories, it enabled us to delve into the subjective depth of each user, highlighting their peculiarities, similarities, and differences. It is important to note that all participants volunteered and did not receive any benefits. Additionally, all participants consented to the use of their data for scientific research purposes, provided anonymity was ensured. We recorded the consents via video and stored them along with the collected data in encrypted spaces.

In Table 1, we schematically describe the main information about the interviewed participants.

**Table 1.** The main information about the participants.

| User | Type of Company | Role | Age |
|------|-----------------|------|-----|
| User 1 | Catering Services Company | Senior Project Manager | 33 |
| User 2 | Public University | Professor—Instructional Expert | 70 |
| User 3 | Multimedia Production Company | Junior Project Manager | 24 |
| User 4 | Multimedia Production Company | Junior Project Manager | 23 |
| User 5 | Multimedia Production Company | Junior Project Manager | 26 |
| User 6 | Multimedia Production Company | Junior Project Manager | 23 |
| User 7 | Event Management Company | Senior Project Manager | 51 |

**Table 1.** *Cont.*

| User | Type of Company | Role | Age |
|------|-----------------|------|-----|
| User 8 | Learning Provider Company | Senior Instructional Designer | 40 |
| User 9 | Multimedia Production Company | Junior Project Manager | 29 |
| User 10 | Event Management Company | Senior Project Manager | 38 |

*3.6. Data Collection*

The ten users were interviewed individually, remotely, and video-recorded using Classic Microsoft Teams software for Window 10 (version 1.6.00.17554). To collect data, we sought a technique that would allow us to observe and detect the user's experience of consuming the educational video in the purest manner possible. We found it in the think-aloud protocol [59,60]. In the first phase of data collection, we asked each of the users to consume the Cyber Gym video and express their inner thoughts verbally during the consumption. The use of the think-aloud protocol, along with the capabilities of the software used, allowed us to record the users' verbal comments, the actions they performed on their screens, and their facial expressions in a synchronized manner.

Before starting the video, we recited the following task to the user:

> Watch the agreed-upon video and do it as if I weren't here. In fact, I'll turn off my microphone and camera. You'll be one-on-one with the video. Remember to verbally express everything you think while you're watching it. I ask you to speak freely and be as sincere and immediate as possible. Whatever crosses your mind, please express it out loud. Once you've finished watching the video, just say, 'I'm done.' Do you find everything clear, or do you have any doubts?

After reciting the task and resolving any doubts, the user shared their screen and commenced watching the video while verbally expressing their thoughts.

In this phase, users performed a "concurrent verbalization" as the information was verbalized while users were considering it. The required type of verbalization was at "Level 2", as the received information was originally encoded in the video medium and needed translation by the user into verbal form. We refer to "Level 2" rather than "Level 3" because, as evident from the task given to the users, we allowed them to freely express whatever came to their minds spontaneously, without directing their focus to specific aspects of the video [59,61].

After watching the video, we conducted semi-structured interviews composed of 11 questions to further explore their experience [62]. In this second phase of data collection, we requested a "Level 3 retrospective verbalization" as users were asked to reflect on the task just performed and verbalize specific aspects of it [59,63]. The exact same questions were posed to all users:

1.  Can you give me a summary of the video you just watched?
2.  What are the things that have impressed you the most positively?
3.  What are the things that have bothered you the most?
4.  Did you find the pace of the video to be fast or slow?
5.  Did the concepts appear clear and thoroughly addressed?
6.  Was the video appropriate for your level of knowledge? Or was it too easy/complex?
7.  Did you get bored during the video?
8.  Did the use of on-screen text help you better understand what was being said?
9.  Is there an image, sequence, or diagram that particularly stood out to you?
10. Do you feel you can apply what you've learned? If so, could you provide an example?
11. Do you have any suggestions for improving the video's quality?

*3.7. Research Procedure*

After agreeing on the meeting time with each participant, the interviewer, i.e., the first author of this paper, sent the user a link to access the Microsoft Teams software room

where the interview would have taken place. Using their own laptop and headphones, the user entered the room and received a briefing on what he or she would be required to do. The interviewer explained that he or she needed to watch the video and speak aloud, expressing inner thoughts verbally. At the end of the video, a brief interview would have followed. While describing the study's objective, the interviewer never revealed the specific interest in the perception of the metaphor but stated that he wanted to investigate the overall video experience. No further details were provided. After this introduction, the interviewer sought the user's consent for video recording. Once the video recording started, the interviewer asked for the user's consent to use the collected data for scientific research purposes. The interviewer also assured the user that sensitive data would not be disclosed, and therefore, user's name would be replaced with the label User 1, User 2, etc. Once the user gave consent, the interviewer sent him or her the link to access the video. Before opening the link, the interviewer recited the task to the user, which was reported in full in the previous paragraph. If the user had any doubts, he or she could ask questions; otherwise, the user could start the video. During the video consumption, the user spoke aloud, expressing his or her thoughts freely. The interviewer did not intervene during this process. At the end of the video, the user said, 'ok, I'm done', and the interviewer turned on his microphone and webcam. Then, the second part of the interview began, where the 11 questions were asked. It should be noted that the same set of questions was posed individually to all ten users in the same order. However, since the interview was semi-structured, some responses were further explored, initiating a genuine conversation between the interviewer and the user. After the interview, the interviewer downloaded the recording in MP4 format and utilized Microsoft Word software for Windows 10 (version 2307) to automatically transcribe it. Subsequently, the interviewer listened to the interview, correcting any errors produced by the automatic transcription software. Once the errors were corrected, the interviewer divided the interview into individual text units and arranged them in a Microsoft Excel spreadsheet (version 2307). Finally, the process of analyzing and categorizing the individual text units began. The described procedure was followed in an identical manner for all ten users. In the next section, we explain how the collected data were analyzed.

*3.8. Data Analysis*

To analyze the collected data, we employed techniques and procedures typical of both descriptive and interpretative phenomenological analysis [49,52,53,57,58] and we adequated them with our research topic, aim and materials [57]. Firstly, we transcribed the ten recorded session into a single Excel sheet and divided them into units. To divide them, we used the criterion of self-consistency, where a unit can be defined as such and only if it makes sense even when taken by itself [52]. For each unit, we conducted five types of categorizations:

1. *Spacetime-based*, where we described the moment and location of the user comment.
2. *Action-based*, where we described the type of action taken and/or facial expressions.
3. *Topic-based*, where we searched keywords referring to gym metaphor.
4. *Approval-based*, where we noted the emergence of an approval.
5. *Disapproval-based*, where we noted the emergence of a disapproval.

Utilizing the topic-based categorization, we were able to filter the collected data to focus on the units related to the gym metaphor. We conducted a second analysis over these units, where we described the structural aspects with which the topic was commented on by the single users [49]. Then, we related the structural aspects to the multiple categorizations performed in the first analysis, and we investigated similarities and differences among the various interpretations provided by the users. Finally, we conducted eidetic reductions [50,53,57]. Through the process of eidetic reduction, which could also be defined as an 'archeology' [55], we brought to light the essential perceptions of the users in relationship to the gym metaphor. We described them, and we interpreted

them. In the results section we will show in detail the essential perceptions and how we reached, described, and interpreted them.

## 4. Results

### 4.1. The impact of the Gym Metaphor on Users' Experience

Drawing from the analysis of user comments, we sought to reach, describe, and interpret the essential perceptions generated by the metaphor in the user's experience.

First and foremost, it should be noted that eight out of ten users commented on the 'gym' metaphor explicitly and spontaneously, without being directly prompted to do so. These eight users did so from the beginning of the video, among their initial comments. In the final interview, at the question 'What are the things that have impressed you the most positively?', six out of these eight users mentioned the gym metaphor. These data reveal that the gym metaphor is an aspect that significantly impacts the user's experience.

To understand how users perceived the metaphor, we present the eight comments that users made at the beginning of the video below (see Table 2).

**Table 2.** The eight comments made by users at the beginning of the video.

| | | |
|---|---|---|
| 1. | User 1: | Alright, the gym metaphor is cute. |
| 2. | User 2: | First of all, we're in a gym, and I didn't understand… Why? [disapproving expression] But then I see it's a training, so maybe this is supposed to be a gym, but… [shakes head]. |
| 3. | User 3: | I like the idea of the gym. |
| 4. | User 4: | Nice training in the gym. |
| 5. | User 5: | Music, cybergym, well, that's cute. Let's enter! [during the video zoom effect]. |
| 6. | User 6: | Regarding the intro, I find the music very engaging, and, personally, I like the context of a training in a gym. |
| 7. | User 7: | The gym setting, I really like the graphics of this video. |
| 8. | User 8: | Right, because we're in the Cyber Gym, so a personal trainer [smiles and nods]. |

Seven out of eight comments refer to the gym in terms and actions of approval: 'I like' (×3); 'cute' (×2); 'nice'; smiles and nods. Only one, number 2, was in terms of disapproval. Furthermore, comments number 2, 4, 6, 8 show that the concept of the gym triggers related concepts: 'training' (×3) and 'personal trainer' (×1).

### 4.2. The Common and Implicit Perception of the Gym

Before analyzing the described effects of the gym metaphor in detail, we put forward the interpretation that there is a common implicit perception underlying these comments, which we summarize in the phrase: "the gym is (as if it was) a physical space in which I am in". Based on this perception, we assert that all comments about the gym derive from it. Now, we describe how we phenomenologically arrived at our interpretation.

Let us observe comments 2, 4, 5, 8. The gym is discussed as if it is actually a space where one is in: "we are in a gym", "let's enter", "we're in the Cyber Gym", "training in the gym". The user making these comments is seated in their chair, yet the gym is configured as a space where it is possible to state actions that indicate physical movements: "being", "entering", "training". A space that metaphorically envelops and in which one metaphorically acts. It is interesting to note User 5's comment and her use of the verb "let's enter". This verb, which alludes to a physical movement, is said in sync with the zoom movement that occurs in the introductory part of video, allowing the user to "enter" the gym (see Figure A1 in Appendix A). The video's movement appears to generate a simulated movement within the user's mind, which is expressed with the word "let's enter". In the final interview, User 5 uses terms related to movement to describe the perception of that moment and links them to the video's zoom effect: "it makes you enter into the situation"; "it catapults you into the situation".

In comments 2, 4, 5, 8, the reference to the "gym" is structured as a space where one can enter, stay, and move. Even in comments 6 and 7, we find the perception of being in something enveloping. The gym is associated with the terms "setting" and "context". These are more abstract terms, but they still refer to something in which we find ourselves and that envelops the entire experience. Instead, the users who recognized the gym as a "metaphor" (User 1) or an "idea" (User 3) highlight the archetypal aspect of the gym, namely, being the principle with which the video has been developed. However, even for them, the metaphor is perceived as a space in which the video experience is enveloped. This can be seen in the comments they make immediately after the ones reported: User 1, "of course, we are in the context of the gym" or User 3, "yes, being in a gym". In all eight users who explicitly mentioned the gym, it is perceived as a metaphor that reifies and structures the video space, a space in which to be and move.

*4.3. Disapprovals and Approvals of the Gym Metaphor*

After describing the spatial–ontological perception of the gym metaphor and the perceived relationship underlying the users' comments, i.e., the gym is as if it was a space in which I am and move, we analyzed the manifestations of approval and disapproval.

Let us start with comment number 2, in which the user expresses his disapproval. The negative judgment reveals a limitation of the metaphor, namely the possibility of not being shared. The user, in fact, understands the metaphor but judges it as incoherent. The user explains his disapproval of the metaphor in these terms:

> "Why is there a personal trainer instead of an IT expert? Why has this situation been recreated, rather than recreating a more familiar and topic-appropriate situation? For example, I receive an email, I have a doubt, I call the IT expert who explains how to recognize phishing". User 2

The user does not immediately see the parallel between the gym and the IT dimension of the content. In addition to not finding it immediate, the user also does not share it culturally; the world of the gym is foreign to his experience. Therefore, the presented situation is perceived as unrealistic, unfamiliar, wrong, and, in one word, incoherent. This generates in the user a strong sense of "estrangement" and "non-involvement", as reported by him in the final interview. In his experience, the metaphor appears as a tool that distances, alienates, and puts him in a closed posture: it is a wall rather than a bridge.

The users who shared the metaphor used aesthetic terms and/or gestures of pleasure to express their approval: "cute" (User 1 and User 5), "nice" (User 4), "I like it" (User 3, User 6, and User 7), and smiling and nodding (User 8). We can interpret the sense of pleasure also as a sense of engaging. In fact, although only one user among the comments above explicitly uses the term "engaging" (User 6), three other users who expressed positive judgments used variations of the term 'engagement' in relation to the metaphor during the final interview:

> "You enter the gym, and you're *involved* in some way with the song, with the questions, with the fact that they're having a dialogue". User 4

> "So, I was in a clear, clean environment that *drew my attention* to those two central figures who were having a dialogue". User 5

> "It [the gym metaphor] *captured my attention*, was an antidote to distraction".
> User 8

We want to understand in more detail why users positively commented on the gym metaphor. The aim is to highlight the key elements that lead to the approval of the metaphor. From the comments collected, we emphasize that one of the main elements of approval lies in the familiarity with the concept of "gym". Four users who appreciated the metaphor have stated that they attend the gym regularly:

> "Let's say that I always go to it [the gym], so, I mean, let's say that the video is foreshadowing a situation..." User 3

"I go to the gym, and I really liked this *mental training* as a metaphor. I liked it a lot. So, *I'm ready to train too*, you know". User 4

"Alright, I like it because I always go to the gym, *let's try some mental gym* and understand what this phishing is". User 5

"The fact that it was set in a gym, the training aspect really appeals to me because it's a passion of mine. So, this idea of *training mentally* as well is very nice". User 6

The possibility to simulate the gym based on real-life experience made the metaphor more engaging, shareable, and comprehensible, allowing for significant parallels to be drawn. For example, three of the comments mentioned above discuss 'mental training'. By this parallel, postures, emotions, and attitudes typical of physical gym training were transferred to the field of cybersecurity training, as further clarified in the following comment:

"I forgot to tell you that I really liked the gym metaphor because it explicitly conveys the idea that it's normal to feel disoriented once you start learning these things. However, the solution isn't just to 'be careful', but to understand that the more you train, the more awareness you gain... Let's say that one becomes competent by looking at more things and examining them from different perspectives. You have to train constantly". User 4

In this comment, the metaphor served the function of orienting the user's attitude: just as we need to constantly train to improve our body, we must also train constantly to enhance our cybersecurity competence.

However, the metaphor does not only orient the user's attitude but also helps orienting themselves within the video they are about to engage with. The metaphorical names that structure the three parts of the video (warm-up, training, cool-down), derived from the world of the gym, allowed users to orient themselves and progress along a familiar linear training path (see Figure A4 in Appendix A). One user expressed it verbally:

"They [warm-up, training, cool-down] give a very clear idea of what we will do later, which is training, as I read above". User 6

The structure of the gym workout is transferred to configure the educational video on cybersecurity. The metaphor shapes the path through a familiar language. The possibility of using familiar language to explain and structure new or complex concepts seems to be an approach that can create meaningful bridges between expert knowledge and the concrete daily experience of the user, as is evident in the following comments:

"So, I'll say something, I like the term 'robust password' placed within the context of the Cyber Gym, one goes to the gym to build muscles, so 'robust password', [...] I liked it, and it sticks in my mind". User 3

"As I already mentioned, I like the association between the gym and some terms used within the course, such as warm-up, training, cool-down [...] so there are associations that, well, they refer to the aspects of the course". User 6

In these two comments, the shared metaphor enables an understanding of the language used to structure the video (warm-up, training, and cool-down) and to structure cybersecurity concepts (the password should be robust).

In addition to structuring concepts and the video, the metaphor also partially structures the way the user thinks. User 8, in his comment upon entering the video, immediately understands the presence of the personal trainer, even though it is not typically associated with the cybersecurity realm. This is a small example of a more extensive phenomenon, which is the ability to understand seemingly unrelated elements through the filter of a metaphor. Being in a gym activates a constellation of related concepts and they are used to understand the video content; in the User 8 case, the role of the personal trainer is comprehended and shared in its educational function because it belongs to the metaphorical world of the gym. In addition to the emergence of related concepts, the metaphor also raises certain expectations in the user. These expectations need to be anticipated by the designer

because, if not fulfilled, they can lead to a perception of incoherence and disappointment. This was the case with the lack of interactivity: users, given the gym metaphor, expected to engage in active actions, not just listen or watch. Some users clearly expressed the need for interactive activities and alternatives to the video, such as quizzes or games.

> "Given how it's presented, and also due to the gym metaphor, I would have expected it to be more interactive. I mean, it wasn't boring, but I felt passive in my learning". User 1

> "I would also include some quizzes, maybe some little games instead". User 3

> "And maybe in between, since the video is short and not too heavy, I would also include some practical games". User 5

The desire for interactivity was also evident in some emblematic actions, such as User 4 pausing the video to answer a question posed by the personal trainer before the trainee did, or User 6 pausing the video to provide her own summary of the seven main rules for recognizing a phishing email. The sense of being in a gym generated an expectation of interactive learning. The presence of only videos disappointed the users.

We also report that two users, User 9 and User 10, who never explicitly and spontaneously talked about the gym, expressed a favorable judgment on the video format.

> "In my opinion, the cartoon reduces the effort of learning, makes it more enjoyable. It had a positive role. I think it makes it more familiar, it reaches you more directly, you feel it as something closer, but above all, it's like you're not studying but pleasantly watching a video, which is something you usually do in your free time, and it doesn't tire you out. I really liked the structure, the introduction of the training, and the conclusion". User 9

> "Also nice is the mode, you know, with the cartoon. I liked it, it makes it very lively, light, so much better than usually, for sure". User 10

In their comments about the video, User 9 states that the chosen mode, the cartoon, helped him familiarize with the content and emphasized that it reaches you more directly. Also, User 10 expressly stated that she liked the chosen mode because it made the video light, lively, and engaging. We did not include these comments in our analysis of the gym metaphor because they do not explicitly refer to it, but we found it useful to report them to demonstrate that the context and mode were also appreciated by these two users, contributing to their engagement.

*4.4. A Schematic Overview of the Results*

We conclude by providing a schematic overview of the conducted analysis. We propose a table summarizing how users commented on and perceived the gym metaphor (see Table 3).

**Table 3.** Overview of the results.

| Type of Comment | User | Example Quotes | Essential User Perceptions |
|---|---|---|---|
| It was spontaneously commented | 8/10 | "First of all, we're in a gym"."Alright, the gym metaphor is nice". | Surprise |
| It was not commented on | 2/10 | — | — |

**Table 3.** *Cont.*

| Type of Comment | User | Example Quotes | Essential User Perceptions |
|---|---|---|---|
| It was commented on in terms of approval | 7/10 | 5/10 — "very engaging, I like the context of a training gym". "You enter the gym, and you're *involved*". | Engagement |
| | | 4/10 — "I like it because I always go to the gym". "I go to the gym, and I really liked this mental training". | Familiarity |
| | | 4/10 — "I really liked the gym metaphor because it explicitly conveys the idea that it's normal to feel disoriented [...] You have to train constantly". "They [the path titles] give a very good idea of what we will do later, which is a training". | Orientation of attitude and orientation in video structure |
| | | 5/10 — "I like the term 'robust password' placed within the context of the Cyber Gym". "We're in the Cyber Gym, so a personal trainer". | Support for concept structuring and activation of related concepts |
| It was commented on in terms of disapproval | 4/10 | 1/10 — "First of all, we're in a gym, and I didn't understand... Why? [...] Why has this situation been recreated, rather than recreating a more familiar and topic-appropriate situation? [...] It estranges me". | Incoherence Unfamiliarity Estrangement |
| | | 3/10 — "Due to the gym metaphor, I would have expected it to be more interactive". | Activation of related concepts IncoherenceDisap- pointment |
| It was commented in term of physical involvement | 8/10 | "We are in a gym". "Let's enter". "Let's train in the gym". | *As if* users were actually inside the gym. |

## 5. Discussion

In the section dedicated to the pedagogical role of the metaphor, we asserted that it can be a tool that helps bridge the gap between the learner and the object of learning, because it allows for breaking down complexity and presenting novelty through simpler and already-familiar concepts [1–5,40]. However, to achieve the educational objective, the metaphor must be understood, shared, and perceived as coherent by the learner [2]. In our phenomenological analysis, we confirmed these assumptions derived from theoretical reflection.

The metaphor, when shared, allows for a closer approach to the concept being learned. Conversely, if perceived as incoherent, it leads to estrangement and non-involvement. In the case of the seven users who positively commented on the gym metaphor, we highlighted that sharing primarily occurs through the familiarity the learner has with the concept used to metaphorize the concept to be learned. In fact, four users who expressed positive opinions are regular gym-goers in their daily lives. This shared familiarity has allowed for creating meaningful connections between the context of the physical gym and cybersecurity learning, such as the constant need for training. Furthermore, this sharing has generated a sense of engagement, verbally expressed by four users. The only user who negatively

commented on the metaphor and perceived it as incoherent verbally expressed a sense of estrangement and non-involvement. The metaphor was criticized, both because the user is unfamiliar with the gym world and because he judged other situations, such as receiving a phishing email at work and consulting an expert, as more coherent. The choice of metaphor and the user's perception of its coherence can generate opposite effects, ranging from engagement to estrangement.

We have also demonstrated that the metaphor can activate a network of concepts already known to the user, which they leverage to understand new concepts. Reformulating the technical language that characterizes cybersecurity through a more familiar language has facilitated conceptualization and orientation within the video. This reformulation did not imply a rejection of logical-analytical thinking but rather an integration between the language of cybersecurity, which can be challenging, dull, and technical, with a metaphorical language to stimulate and bring the user closer to complex concepts [1,4]. One user explicitly stated that the use of metaphorical adjectives, such as "robust" for the analytical definition of the password, helped fix the concept in her mind. Another user asserted that the use of metaphorical terms such as "warm-up", "training", and "cool-down" to express the didactic functions of the three parts of the video helped her recognize and orient herself in the instructional structure.

As stated by Lakoff and Johnson [20] and Sfard [1], every metaphor tends to illuminate certain aspects while obscuring others. Choosing to set cybersecurity training within a gym has illuminated multiple concepts, such as the appropriate mindset to adopt, but has obscured other aspects, such as the real situation where one receives a phishing email and contacts an expert. The negative comment from User 2 highlighted this omission. The metaphor also illuminated and activated a series of expectations that, if not met, can generate a sense of dissatisfaction. As described, three users verbally expressed expecting moments of interactivity because the gym metaphor ignited the perception of active training. The presence of only videos and the absence of interactions disappointed this expectation activated by the metaphorical concept of a gym. If Sfard [1] asserts that a metaphor alone is not enough to express a complex concept, we add that even a single medium is not sufficient. In the case of teaching and learning cybersecurity, the analyzed metaphorical video is only one element that must be integrated into a network of other educational contents through which the users can learn in a multimodal, multimedia, and multimetaphorical manner the competence they are training for.

Furthermore, we report that no user perceived the use of the metaphor as cognitive overload, confirming the results of the study by Moreno and Mayer [40]. Reflecting on the collected data, we affirm that the metaphor can serve cognitive aid functions. We summarize them in the three essential functions attributed to metaphor by Lakoff and Johnson [20]:

- Ontological functions, as the metaphor reified the space of education, configuring it as a space to be in and move within;
- Orientating functions, as the metaphor oriented the user's attitude and their movement while navigating the video;
- Structuring functions, as the metaphor allowed for the illumination and activation of a series of related concepts to anticipate and understand the structure of concepts and video.

From the perspective of the relationship between metaphor and educational video, we would like to highlight three main aspects that seem to be absent from the scientific literature. The first is the surprise effect: the fact that eight users spoke about the gym metaphor spontaneously and from the beginning shows that it is unusual for an educational video to be set in a metaphorical space. This surprise effect is important, in our opinion, because it transforms into a curiosity effect that draws the user in and stimulates their attention.

The second aspect concerns the visual role of the metaphor and the medial potential of the video. Although it has been partially highlighted by Moreno and Mayer [40], even

if not in reference to the video format, in the scientific literature, the metaphor is mainly addressed as a verbal tool that the teacher can use at certain times to clarify and simplify this or that concept expressed in the video lesson [5,29,30]. Spoken or written metaphors can activate a meaningful image in the mind of the listener, but the multimedia format, specifically the video, can visually show the metaphorical image that is created when two seemingly distinct concepts are combined. In this specific case, we believe that saying only in words, 'Cybersecurity learning is similar to training in the gym because it presupposes constant training', is quite different and less effective than situating cybersecurity training inside a gym visually.

The third aspect is connected to the second since it concerns again the visual role of the metaphor and the medial potential of the video. In fact, videos allow us to create a metaphorical setting in which to situate and articulate the concepts to be learned. The possibility of situating the object of learning within a metaphorical space, in addition to surprising the user and structuring the concepts, strengthens the process of embodied simulation, described in recent neuroscience research in relation to the video experience [46]. We have described that users naturally claim to be inside the gym or moving within it, despite being seated in their chairs. This paradoxical naturalness can be explained through the natural phenomenon of embodied simulation, allowing users to perceive themselves as if they were truly inside the gym. A metaphorical setting makes the video more immersive, reinforces simulation, and allows the user to personally enter into it.

## 6. Conclusions

In our study, we investigated how users perceived the use of metaphor in the educational video "Cyber Gym" to understand whether metaphors aid or hinder the learning process through videos. The phenomenological results revealed that using a metaphor to contextualize and describe the object of learning impacts users' experiences and can assist in simplifying complexity through more familiar and straightforward concepts, thereby facilitating educational mediation processes. However, this facilitation occurs only when the metaphor is perceived as coherent by users. In fact, when shared and judged as coherent, metaphor fosters engagement and connection, but if perceived as incoherent, it leads to estrangement and non-involvment. Familiarity with the concept used to create the metaphor is a fundamental element through which users judge coherence and comprehend the object of learning, because it helps users to activate a meaningful network of related concepts and expectations to break down complexity and understand novelty. Therefore, when using metaphors to teach and learn new and complex concepts, it is essential for users to be familiar with and perceive the metaphor as coherent with the content. From the scientific literature, the use of metaphors in educational videos is typically confined to the verbal-discursive aspect employed by the speaker-teacher. In contrast, our study demonstrates that metaphor in educational videos can be a multimodal tool that facilitates mediation, creates metaphorical settings, and generates a surprise effect leading to engagement and curiosity. In fact, we showed that videos have the capability to present the metaphor not only verbally but also visually, strengthening the metaphorical representation of the concept to be learned in a multimodal way. Moreover, videos can create metaphorical settings to situate the content of learning. This possibility reinforces embodied simulation processes activated during video consumption and allows users to place themselves within the metaphorical setting, fostering a heightened sense of engagement and participation.

However, we acknowledge that our study has several limitations. We highlight the three main ones:

- The study focused only on one metaphorical video, not considering other examples;
- The study focused on a small and non-representative sample;
- The study focused solely on the metaphor and not on other elements that constitute the video.

For the future, we are already working to overcome the third limitation by integrating the analysis conducted in this study with other elements, such as the presence of characters

and the use of simulated dialogue. We have noticed that comments related to the gym often accompany remarks about the characters inhabiting it and their dialogic interaction. Furthermore, we are observing that the simulated perception of being inside the gym leads users toward a particular relationship with the characters, to the extent that they perceive characters as real, and to an intense participation in the dialogue, to the extent that they perceive themselves as active characters. Subsequent studies will focus on how users have perceived the characters and their dialogic interactions within the metaphorical setting.

**Author Contributions:** Both authors contributed to the study's conception and design. M.N. conducted the material preparation, data collection, and analysis. The initial draft of the manuscript was written by M.N. and L.D. provided comments on previous versions. All authors have read and agreed to the published version of the manuscript.

**Funding:** This research received no external funding.

**Institutional Review Board Statement:** The study was conducted on a sample of informed subjects in accordance with Article 13 of EU Regulation 679/2016. The data of the individuals are processed for the purpose of managing the sample itself and do not fall among the published information. Data collection was carried out with prior information and authorization for the use of information from the interviewed sample, in accordance with GDPR provisions. The interviewed sample was properly informed about the purposes of the research and the interview itself, as well as the use of information/opinions expressed during the interview. I, the undersigned Michele Norscini, the author of the research, have a recorded video copy of the authorization for the use of information provided by the interviewees and the corresponding authorization for the use of the data.

**Informed Consent Statement:** All participants confirmed that their participation in the study was voluntary and agreed that the data provided could be used in the research on an anonymous basis.

**Data Availability Statement:** The data presented in this study are uploaded to the following cloud space: data_manuscript_metaphor_googledrive. Anyone interested can access it upon request from the corresponding author, in accordance with the interests of the University of Macerata and the University of Latvia.

**Acknowledgments:** The authors acknowledge the assistance provided by Skilla company in making the video available and reaching potential users for interviews. The authors express gratitude to the users who participated in the research; without their involvement, this work would not have been possible.

**Conflicts of Interest:** The author Michele Norscini served in the instructional design team of the Cyber Gym video. The authors have no relevant financial or non-financial interests to disclose.

## Appendix A

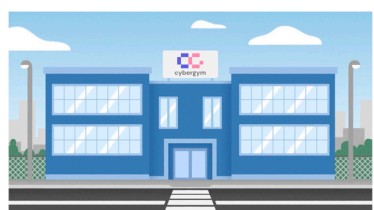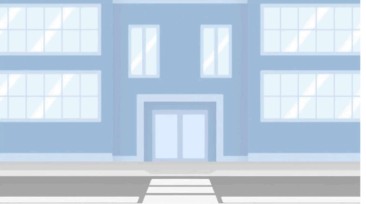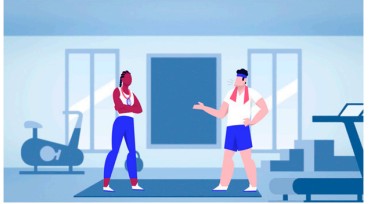

**Figure A1.** The zoom movement with which the user enters the gym.

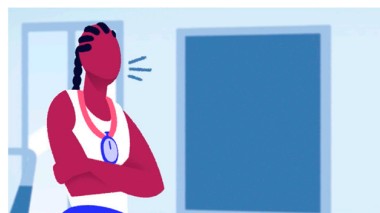

**Figure A2.** The personal trainer character.

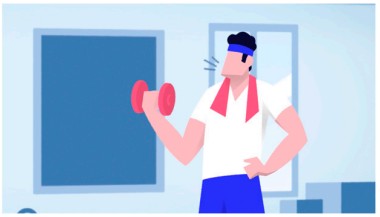

**Figure A3.** The trainee character.

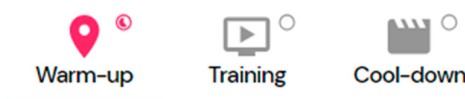

**Figure A4.** The structure of the video.

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
