# Peer review of "Metaphors in Educational Videos"

_education, doi:10.3390/educsci14020177_

Round 1

Reviewer 1 Report

Comments and Suggestions for Authors

The manuscript Metaphors in Educational Videos provides an important and interesting insight into using metaphors in educational videos. However, the manuscript could be improved regarding the following minor points:

It would be useful if the use of metaphors were defined right from the beginning in the Introduction.

A more in-depth review of the subject could be provided within the literature review. This could include definitions and use of metaphors within different disciplines, such as in literature and poetry as well as maybe art therapy and the arts.

Some sentences need clarification, such as on p. 4:

With metaphorical educational video, we refer to a specific type in which the knowledge to be learned is visually situated in a metaphorical space and it is multimodally expressed in light of the chosen metaphor.

What is referred to as a “metaphorical space”? What does “multimodally expressed in light of the chosen metaphor” mean? I suggest further clarifications.

I question if this general information about education is needed:

“If metaphor proves to be an indispensable tool for cognition and language, it can be consciously used to support the phenomenon of educational mediation. As Cameron (2002) observes, echoing Vygotsky's reflection, the goal of teaching and learning is to bridge the gap between the learner and the object of learning. To achieve this, the teacher transposes the savoir-savant (Chevallard, 1985) or scientific concepts (Cameron, 2002) into instructional knowledge and then mediate it to the learner through various modalities, such as a classroom lecture or an educational video.”

I found too much of the text to be within quotation marks. The text would flow better and be easier to read if the references were integrated into the text without being within direct quotes:

Through such metaphorical expressions, scientific theories are constructed, and as Scheffler (1991, p. 45) states, "there is no obvious point at which we may say, 'Here the metaphors stop and the theories begin.'"

"… two domains that are distinct and somehow incongruous, but whose juxtaposition can be made sense of" (Cameron, 2002, p. 674).

Scheffler (1991, p. 45) states: "there is no obvious point at which we may say, 'Here the metaphors stop and the theories begin.'"

In the process of didactic transposition and mediation, metaphor can be used "pedagogically as a kind of stepping stone in the reduction of conceptual alterity, or to prompt conceptual restructuring" (Cameron 2002, p. 676).

and the definition by Ibrahim and colleagues (2012) is provided, which describes video as "a format of presenting information as a stream of dynamic visual and auditory content" (p. 156).

Essentially, what distinguishes an educational video from other types is the presence of an "explicit learning goal" and an "intent to teach" (Fyfield et al., 2022, p. 156).

For example, the study by Alnajjar et al. (2022) presents "a method for detecting metaphors in the new dataset based on the textual content of the videos" (p. 24).

Comments on the Quality of English Language

Moderate editing of English language required

Reviewer 2 Report

Comments and Suggestions for Authors

I find the phenomenological procedures followed to be refreshing. Although the qualitative approach is not particularly strong, the insights given by the authors still make it a welcome addition to the literature. That being said, I do have a few remarks. Giving us more dynamic/interactive insight into the game would help a lot.

·        In "Appendix" there are some frames of the video; why not link to complete video or a walkthrough of the video.

·        it would be usefull to report the 'level' of think aloud you are using, it seems like level 3 think aloud and naturally reference the literature. The think-aloud can also be more detailed. What is retrospective think-aloud, level 3, or what specifically?

·        I would love to know the ages of the participants. perhaps even describe the type of company or use any other way to discern the employees' experience or expertise.

·        Because the data collection states the microphone and camera are turned off, I assume the procedure was run on a computer or laptop, maybe a tablet? Please specify. Were they wearing headphones? Act as if I wanted to reproduce this. It would also be useful to have version numbers of the software and references. Not all versions of Word transcribe in the same manner. 

Reviewer 3 Report

Comments and Suggestions for Authors
  1. Content Contextualization: The manuscript effectively contextualizes its content within the existing theoretical and empirical research on metaphors in education. It references relevant historical and current perspectives, situating its contribution within this broader discourse. 
  2. Relevance of Cited References: The references cited in the manuscript are pertinent and support the research. They encompass foundational theories and recent studies, indicating thorough engagement with relevant literature.
  3. Clarity of Research Design and Methods: The research design, questions, and methods are clearly articulated. The phenomenological approach, participants, data collection, and analysis procedures are presented, providing a clear understanding of the research process. To enrich the metholody section, consider to further clarify the rationale behind choosing the phenomenological approach and its suitability for this research. Additionally, it would be beneficial to elaborate on the selection criteria for participants to enhance the rigorousness of the study's scope, especially how this study avoided the Self-Selection Bias while the non-representative convenience sampling technique had been used.
  4. Coherence and Balance in Arguments and Discussion: The arguments and discussion of findings are coherent and balanced. The manuscript presents a compelling narrative, linking the empirical findings to broader theoretical implications in a logical manner.
  5. Presentation of Empirical Results: The results of the empirical research are clearly presented. The manuscript effectively utilizes qualitative data to illustrate key findings, enhancing reader comprehension. Improve the organization of results for better readability through thematic categorization in the result section. Finally, in the discussion section, to better contribute to the academic community's knowledge, please consider expanding the discussion on comparing the study's findings and comparative analysis with similar studies if applicable. 
  6. Adequacy of References: The article is well-referenced, with a broad range of sources substantiating the research. The references are appropriate and contribute to the academic rigour of the manuscript.
  7. Support for Conclusions: The conclusions drawn in the manuscript are well-supported by both the results presented and secondary literature. The study's findings are effectively linked to its broader implications and potential impacts on the field.
